# Global analyses of biosynthetic gene clusters in phytobiomes reveal strong phylogenetic conservation of terpenes and aryl polyenes

Arijit Mukherjee,[1,2] Hitesh Tikariha,[1,2] Aditya Bandla,[2,3] Shruti Pavagadhi,[2,3] Sanjay Swarup[1,2,3]

**ABSTRACT**   There are gaps in our understandings on how did the evolutionary relationships among members of the phytobiomes shape their ability to produce tremendously complex specialized metabolites under the influence of plant host. To determine these relationships, we investigated the phylogenetic conservation of biosynthetic gene clusters (BGCs) on a global collection of 4,519 high-quality and nonredundant (out of 12,181) bacterial isolates and metagenome-assembled genomes from 47 different plant hosts and soil, by adopting three independent phylogenomic approaches (*D*-test, Pagel's λ, and consenTRAIT). We report that the BGCs are phylogenetically conserved to varying strengths and depths in their different classes. We show that the ability to produce specialized metabolites qualifies as a complex trait, and the depth of conservation is equivalent to ecologically relevant complex microbial traits. Interestingly, terpene and aryl polyene BGCs had the strongest phylogenetic conservation in the phytobiomes, but not in the soil microbiomes. Furthermore, we showed that terpenes are largely uncharacterized in phytobiomes and pinpointed specific clades that harbor potentially novel terpenes. Taken together, this study sheds light on the evolution of specialized metabolites' biosynthesis potential in phytobiomes under the influence of plant hosts and presents strategies to rationally guide the discovery of potentially novel classes of metabolites.

**IMPORTANCE**   This study expands our understandings of the biosynthetic potential of phytobiomes by using such worldwide and extensive collection of microbiomes from plants and soil. Apart from providing such vital resource for the plant microbiome researchers, this study provides fundamental insights into the evolution of biosynthetic gene clusters (BGCs) in phytobiomes under the influence of plant host. Specifically, we report that the strength of phylogenetic conservation in microbiomes varies for different classes of BGCs and is influenced as a result of plant host association. Furthermore, our results indicate that biosynthetic potential of specialized metabolites is deeply conserved equivalent to other complex and ecologically relevant microbial traits. Finally, for the most conserved class of specialized metabolites (terpenes), we identified clades harboring potentially novel class of molecules. Future studies could focus on plant–microbe coevolution and interactions through specialized metabolites building upon these findings.

**KEYWORDS**   biosynthetic gene clusters, phylogenetic conservation, plant microbiome, squalene hopene cyclase, squalene phytoene synthase, terpenes

Address correspondence to Sanjay Swarup, sanjay@nus.edu.sg.

Arijit Mukherjee and Hitesh Tikariha contributed equally to this article. Author order was decided alphabetically.

The authors declare no conflict of interest.

See the funding table on p. 13.

Plant-associated microbiomes, or phytobiomes, provide critical services for plant growth, development, and health (1–3). Host plants and their phytobiomes collectively produce diverse specialized metabolites of immense importance, influencing

the chemical ecology of different niches (4, 5). Importantly, these metabolites shape the outcome of "microbe–microbe" and "plant–microbe" interactions. Specific classes of molecules, such as benzoxazinoids, phenylpropanoids, and terpenes secreted from plant roots facilitate root microbiome assembly (5–8). Similarly, terpenes, benzenoids, and methanol from leaf exudates shape leaf microbiomes (9). Moreover, specialized microbial metabolites also modulate plant growth and defense (10, 11).

The specialized metabolites from microbes are encoded by numerous families of biosynthetic gene clusters (BGCs) (12). BGCs are highly complex in terms of their genetic organization, as they comprise enzymes, regulators, and transporters (13, 14). They are grouped into seven major classes based on their chemical structures, which are further grouped into subcategories based on both specific chemical moieties and gene cluster similarity (15–17). For example, the terpene class consists of the subcategories hemiterpenes, triterpenes, and tetraterpenes, which also have correspondingly different BGCs (18). On the other hand, structurally conserved metabolites such as aryl polyenes do not have any evident subdivision (19). Such complexity of BGCs limits our understanding of their influence on the "chemical ecology" with respect to phytobiome function.

Phylogenomic approaches, such as phylogenetic trait analysis, can be used to determine the consensus between microbial clades and their associated traits (20). Phylogenetic conservation of specific traits has been used in classical ecology (21), and now, with refined tools, it can be applied in microbial ecology (22). Recently, by the use of this approach, phylogenetic conservation of carbon assimilation (23), responses to nitrogen addition (24), and several other functional traits (22) in microbes have been well characterized. Since specialized metabolites directly govern plant–microbe and microbe–microbe interactions, the ability to produce these metabolites can also be considered an "*effect trait*" (20). Therefore, this approach can be adopted to investigate the association of specific categories of specialized metabolites with microbial clades within and between different microbial ecosystems.

Here, we investigated how phylogenetic relationships of microbial members within and between plant and soil ecosystems have shaped their potential of secondary metabolite biosynthesis. First, we collated phytobiome data sets from cultured bacterial isolates (referred as "plant isolates") and metagenome-assembled genomes (referred as "plant MAGs") and included soil-associated cultured bacterial isolates (referred as "soil isolates") and MAGs (referred as "soil MAGs") as reference. Next, we asked the following questions: (i) Do phylogenetically related microbial members have similar secondary metabolite biosynthesis potential in phytobiomes? (ii) If so, what are the strength and depth of the phylogenetic conservation of different BGC classes among phytobiomes compared to the soil microbiomes? and (iii) How could this approach be used as a tool to identify novel microbial clades that are associated with different subgroups of specialized metabolites? To address these questions, we first predicted the BGCs of 4,519 high-quality and nonredundant genomes (both isolates and MAGs from plant and soil) and investigated their phylogenetic distribution. Next, we calculated the strength and depth of the phylogenetic conservation of BGCs of the four data sets by the use of three independent statistical approaches. Finally, for the most conserved BGC class in the plant isolates, we applied sequence-based analyses to identify microbial clades harboring potentially novel secondary metabolites. Overall, these approaches led to the identification of the patterns of phylogenetic conservation of BGCs in plant and soil microbiomes and have potential application in the guided discovery of novel specialized metabolites.

## MATERIALS AND METHODS

### Study inclusion

The genome sequences of plant-derived isolates were searched and retrieved from two main sources: the JGI (Joint Genome Institute—https://jgi.doe.gov/) and PATRIC databases (25). We followed the following criteria to select for plant-derived isolates:

JGI: filtering criteria—Analysis Project → Genome Analysis (Isolate), Study Ecosystem → Host-associated, Project, Sequencing Strategy → Whole Genome Sequencing, Study Ecosystem Category → Plants; PATRIC: filtering criteria—Filtered by plant name, excluded plasmid and poor-quality genomes (data included until 31 December 2020). Isolates for which either the plant host name was missing or niche information was unavailable were removed. Finally, we merged the plant-derived isolates after removing duplicated (using assembly number) and poor-quality genomes (>5% contamination or <95% complete). This resulted in 4,931 genomes of plant-derived isolates (Table S2). Similarly, genome sequence of soil isolates was retrieved from JGI: filtering criteria—Sequencing strategy → Whole Genome Sequencing, Study Ecosystem Category → Terrestrial; Habitat → soil. Source of isolates, such as glaciers, greenhouse, or having possible plant influence, was excluded. Finally, we merged the soil isolates after removing duplicated (using assembly number) and poor-quality genomes (>5% contamination or <95% complete). This resulted in 2,572 soil isolates.

The genome FASTA files were then downloaded via their genome assembly number using Bioinformatics Tools (bit) (https://github.com/AstrobioMike/bit#bioinformatics-tools-bit) (26). Missing metadata were added manually [located from the NCBI database and the literature, genome size using Quast (v5.0.2) (27), and completeness using CheckM (v1.2.0) (28)]. Taxonomic assignment of the genomes was performed using GTDB-Tk v1.5.1 (29). Furthermore, all the genomes were dereplicated at 98% ANI (30) with fraction alignment of 0.6 using dREP v2.6.2 (31). This cutoff was chosen based on a recent benchmark that demonstrated 98% ANI as an appropriate threshold for generating sub-species level representation (30) of genomes, and in our case, it best represented nonredundant genomes while preserving the microheterogeneity of BGCs. Finally, the resulting 1,395 genomes from plant isolates and 1,768 soil isolates were used for further analyses.

Plant and soil metagenome-assembled genomes (MAGs) were considered from the Earth Microbiome project (32) and our recently published study (33). First, we filtered MAGs belonging to "terrestrial ecosystem" and "soil ecosystem" types to consider them as soil MAGs. Furthermore, this list of soil MAGs was manually curated to remove MAGs derived from the rhizosphere to exclude plant influence. For plant MAGs, we collated the MAGs from both the Earth Microbiome data set and the data set provided by Bandla et al. (33). Bulk and compost soil MAGs from Bandla et al. were excluded from the list of plant MAGs. Both data sets were curated to include only high-quality (≥70% completeness and ≤5% contamination) MAGs. Furthermore, the MAGs in each data set were dereplicated using the same criteria (as used for isolates), 98% ANI with a fraction alignment of 0.6, resulting in 573 plant MAGs and 783 soil MAGs.

## BGC prediction

We used antiSMASH (v5.1.2) (34) (Command: antismash --genefinding-tool prodigal --asf --cb-general --cb-subclusters --cb-knownclusters –pfam2go) for BGC predictions. The antiSMASH results were processed using BiG-SCAPE (v1.0.1; using GCF or Gene Cluster Family clustering threshold of 0.3) (35) and MIBiG (15) reference database version 1.4 for clustering the sequences into systematic BGC categories. The BGCs were classified into 11 classes as per the output from BiG-SCAPE (and as per their abundance in the whole data set, some were moved into and out from class Others), i.e., nonribosomal peptides (NRPS), ribosomally synthesized and posttranslationally modified peptides (RiPPs), terpenes, aryl polyenes, beta-lactones, hserlactone, siderophores, PKS_NRP_hybrids, polyketide synthase other (PKS_other), PKSI, and others. The output table was parsed to obtain the final table of the number of BGCs in each class for each genome across all four data sets.

## Phylogenetic tree and BGC distribution

An unrooted phylogenetic tree of both isolates and MAGs was constructed separately using the GTDB-Tk v1.5.1 *de novo* option, which uses FastTree (36) method for tree

construction. To obtain a more robust maximum likelihood (ML) tree, the supermatrix of all the sequences for each data set from GTDB-Tk output was trimmed separately using BMGE (Block Mapping and Gathering with Entropy) v1.12 (37). The trimmed alignment was used to reconstruct the best consensus tree by IQ-TREE 2.2.0.3 (38) with Q.yeast+F+R10 as substitution model (determined by model finder), branch support calculated by SH-like approximate likelihood ratio test with 1,000 bootstrap replicates along with 1,000 ultrafast bootstrap approximation. The phylogenetic tree for plant isolates was rooted at the Desulfobacterota phylum. For plant MAGs, soil isolates, and soil MAGs, the trees were rooted at the Phylum Cyanobacteriota. The BGC count and metadata were overlaid over the phylogenetic tree using iTOL (39) to visualize the BGC distribution.

The phylogenetic distance among plant isolate members was calculated based on the "cophenetic distance," whereas BGC dissimilarity was calculated based on the "Bray-Curtis" (40) distance from their BGC count table. The strength (Mantel's $r$) and significance ($P$ value based on 1,000 permutations) of the correlation between these two-distance matrices were calculated based on Mantel's test using the "ade4" package in R (41).

Principal coordinate analysis (PCoA) was performed based on "Bray-Curtis" distance from their BGC count table. Statistical significance was calculated based on PERMANOVA using the "vegan" package in R (42).

We estimated the differential enrichment of BGCs among taxa by modeling the count of BGCs in each genome. The "negative binomial" distribution was fitted to model the count of BGCs against genus. Model selection was performed based on residual diagnostics and comparing the Akaike information criteria of the models using the "DHARMA" package in R (43). Differential enrichment for BGCs has been shown for the top 7, and only significantly abundant genera in isolates (plant and soil) data sets.

## Phylogenetic conservation of BGCs

Phylogenetic conservation of BGCs was inferred based on three independent approaches: the $D$-test of Fritz and Purvis (21), lambda ($\lambda$) statistics of Pagel (44), and consenTRAIT of Martiny et al. (22). Notably, these approaches are different in principle. The $D$-test calculates the strength of phylogenetic conservation from binary traits, whereas Pagel's $\lambda$ calculates phylogenetic signal from continuous traits. On the other hand, consenTRAIT calculates the genetic depth of conservation from binary traits. Each of these approaches has its own advantages and limitations; for example, consenTRAIT considers the clades with mixed responses, in contrast to $D$-test (24). However, both consenTRAIT and $D$-test/Pagel's $\lambda$ calculate different attributes of phylogenetic conservation—genetic depth and strength of phylogenetic conservation, respectively. The raw count table of BGCs was converted to a binary table by setting counts greater than or equal to 1 to 1.

D-statistics for phylogenetic conservation were calculated based on binary count table of BGCs, using the default parameters of phylo.D function ("caper" package in R) (45). We permuted the tips of the tree 1,000 times based on a random evolution model and Brownian mode (BM) of evolution to estimate the statistical significance. D-statistics with $P_{random} <0.05$ and $P_{Brownian} <0.05$ were considered statistically significant. Pagel's $\lambda$ for continuous trait values were calculated using the "phylosig" function of "geiger" package (46) in R with default parameters. Statistical significance was calculated based on likelihood ratio test with random evolution model ($\lambda = 0$). We performed consenTRAIT to determine the genetic depth at which the traits are conserved. consenTRAIT finds clades at which 90% of the descendants have the trait of interest (here, each of the BGCs) and calculates the mean genetic depth ($\tau_D$) of those clades. We used the getTraitDepth function from the "castor" package in R, and the parameters used were as follows: min_fraction = 0.9, count_singletons = TRUE, singleton_resolution = 0, weighted = FALSE, as described by Martiny et al. (22). Statistical significance was determined by permuting the tree tips 1,000 times and when the proportion of times $\tau_D$ was less than the observed

**TABLE 1** Mean genetic depth of conservation of individual BGC classes among the four data sets[a]

| BGC category | Plant isolates | Soil isolates | Plant MAGs | Soil MAGs |
|---|---|---|---|---|
| Aryl polyene | **0.02** | 0.021 | 0.04 | 0.049 |
| Beta-lactone | **0.018** | **0.031** | **0.052** | 0.051 |
| Hserlactone | **0.023** | 0.027 | 0.038 | 0.044 |
| NRPS | **0.026** | 0.036 | 0.051 | 0.057 |
| RiPPs | **0.032** | **0.041** | **0.061** | 0.061 |
| Siderophore | **0.018** | 0.028 | 0.043 | 0.037 |
| Terpenes | **0.036** | **0.05** | **0.069** | **0.07** |
| PKS_NRP_Hybrids | **0.022** | 0.027 | 0.054 | 0.054 |
| PKS_other | **0.022** | **0.034** | 0.039 | 0.056 |
| PKSI | **0.016** | 0.029 | **0.067** | 0.059 |
| Others | **0.019** | 0.033 | **0.06** | 0.05 |
| Average | 0.0229 | 0.0325 | 0.0522 | 0.0535 |

[a]Bold numbers indicate statistically significant mean genetic depth ($\tau_D$) of conservation ($P < 0.05$).

$\tau_D$. The mean, maximum, and minimum genetic depths for each of the BGCs were subsequently calculated and shown in Table 1.

## Terpene network and domain analysis

We investigated the amino acid sequence similarity network of only terpenes in phytobiomes (plant isolates and plant MAGs) since terpenes were found to be more abundant and diverse than aryl polyenes. The network file of terpenes containing GCF information of both plant isolates and plant MAGs obtained from BiG-SCAPE was overlaid with taxonomic information (at the class level) of the genomes. The network was then visualized using Cytoscape (v3.8.2) (47). The nodes were colored according to the bacterial class and MiBIGs referenced BGCs.

The Pfam domain list belonging to terpenes was extracted from the BiG-SCAPE output and was sorted based on the number of hits. Among all the domains, only those belonging to core biosynthetic genes and with high abundance were considered for further analysis. The presence/absence data of these domains were used to visualize their distribution in the phylogenetic tree of the plant isolates. To identify differences in the sequences with SQS/PSY (squalene/phytoene synthase) domain, the sequences were extracted from the BiG-SCAPE output and aligned using MAFFT (v7.475) (48). Since more than 80% of gaps were found after alignment, BMGE (v1.12) (37) was used to extract regions from the multiple sequence alignment for phylogenetic inference. The BMGE output was then used for phylogenetic tree construction by RAxML (v8.2.12) (49) with the following parameters: GAMMA model of rate heterogeneity and maximum likelihood (ML) estimate of alpha-parameter with 100 bootstraps. The consensus bootstrap tree was generated using the majority consensus option in RAxML. The resulting tree was then visualized using iTOL. The same method used for Pfam domains was applied to extract, align, and visualize SQS-PSY domains.

## RESULTS

### Phylogenetic distribution of BGCs in phytobiomes

We mined publicly available databases to generate a comprehensive catalog of isolates and MAGs from plants and soil microorganisms (Tables S2, S5, S8, and S11). Taken together, this collection represented a total of 4,931 plant isolates, 1,523 plant MAGs, 2,572 soil isolates, and 1,316 soil MAGs after manual curation and confirmation of the source of isolation from the plant or soil environment (see Materials and Methods). Our collection of phytobiomes is broad, covering 47 different plant hosts across 90 countries (Fig. S1A and C; Tables S2 and S5). The quality profiles of all the data sets, including genome size, completeness, and contamination, are shown in Fig. S1B. The genomes of plant and soil isolates were larger and near complete (median completeness

= 99.59% and 99.54%, respectively; median genome size = 5.74 Mb and 6.16 Mb, respectively) in comparison with those of the plant and soil MAGs (median completeness = 82.93% and 88.81%, respectively; median genome size = 3.69 Mb and 3.49 Mb, respectively). Our collection of phytobiomes covers the most abundant phyla (Proteobacteria, Actinobacteria, Firmicutes, and Bacteroidetes at a ratio of 66:18:9:5) in plant environments, as suggested in previous studies (50, 51). Notably, Firmicutes were underrepresented in MAGs compared to the isolates in both plant and soil (Fig. S1D). However, the collection of plant MAGs covered the most abundant taxa found in the plant environment.

Considering only high-quality and nonredundant genomes, we report a total of 12,916, 23,152, 2,318, and 2,307 predicted BGCs from the plant isolates, soil isolates, plant MAG, and soil MAG data sets, respectively (Tables S3, S12, S6, and S9). NRPS was found to be the most abundant BGC, with an average BGC counts per genome of 2.047, 2.959, 0.916, and 0.706 for plant isolates, soil isolates, plant MAGs, and soil MAGs, respectively (Tables S3, S12, S6, and S9). The number of total BGCs per genome varied considerably among members (Fig. 1A) and among the four data sets (Fig. S1E). We observed characteristic patterns in the distribution of different BGC classes among taxonomic members of plant isolates (Fig. 1). Visualization of the distribution of BGCs in plant isolates revealed that members of the class Actinomycetes (*Streptomyces* sp.) encode a higher number of BGCs per genome compared to those of other taxa and represent a metabolically dynamic clade (Fig. 1A and inset A-1). This was also reflected in the BGC profiles of soil isolates (Fig. 2, inset A-1). Moreover, members of the class Alphaproteobacteria (such as *Mesorhizobium*, *Bradyrhizobium*, and *Rhizobium*) encoded a higher number of homoserine-lactones compared to other members of their taxonomic group (Fig. 1A and inset A-2). However, this trend was particularly absent in soil isolates, suggesting differential patterns of BGC distribution in soil and plant isolates (Fig. 2 and inset A-2). Additionally, the principal coordinate analysis (PCoA) of the BGC profile showed that members of the most abundant taxonomic groups (top 7 abundant genera in plant isolates) occupy distinct biosynthetic space in the coordinate plot, suggesting that BGCs are phylogenetically clustered (Fig. 1B). Similarly, the biosynthetic space of soil isolates also displayed a distinct clustering in the coordinate plot (Fig. 2B).

Differential enrichment analysis revealed a parallel trend of BGC distribution, as observed in Fig. 1A. For example, *Streptomyces* (phylum Actinomycetes) encode higher number of BGCs involving NRPS, PKS-other, PKS-NRP hybrids, siderophores, and terpenes in their genomes (negative binomial; $P < 0.05$ and Fig. 1C) compared to those of other taxa. We noted that members of the *Bacillus_A* genus possess equal proportions of NRPS and RiPP BGCs but have a lower number of BGCs involving aryl polyenes and PKS-other in their genomes compared to those of other taxa (Fig. 1C). Whereas this trend was not particularly reflected in soil isolates, suggesting that certain BGCs are differentially distributed in phytobiomes (Fig. 2C). The detailed results for differential analyses of BGCs for plant and soil isolates are presented in Tables S15 and 16, respectively.

## The strength and depth of phylogenetic conservation vary for different BGCs among phytobiomes

Since we observed that the biosynthetic space of plant isolates is taxonomically distinct, we next tested whether there is a relationship between their phylogenetic distance and their BGC profiles using dissimilarity measures. Overall, there was a moderate yet significant correlation between BGC profile dissimilarity and phylogenetic distance (Mantel's $r = 0.31$; $P = 0.0009$; Table S14), suggesting that BGCs may be phylogenetically conserved. To investigate this further, we considered the individual BGC classes and sought to determine what the strength and depth of their phylogenetic conservation were among the phytobiome clades. We first investigated phylogenetic conservation in plant isolates and then used plant MAGs as an independent data set for validation. Next, we compared this pattern of phylogenetic conservation of BGCs with that of the soil isolates and soil MAGs separately, to understand how their conservation differs between

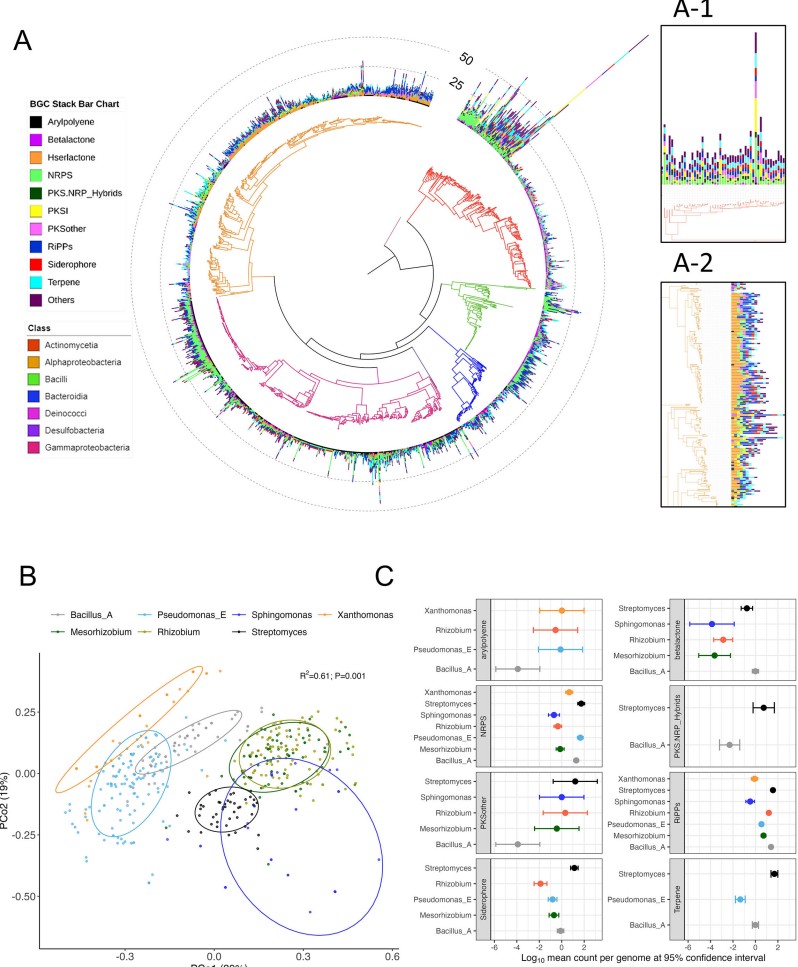

**FIG 1** Phylogenetic distribution of BGCs in plant isolates. (A) Phylogenetic tree displaying distribution of BGCs among members of plant isolates. Stacked bars in the outer rings indicate the count of different BGCs (shown in different colors) encoded by individual members. Taxonomic information (at the class level) is shown in colors of the phylogenetic tree branches. Inset A-1 highlights that Actinomycetes phylum possess overall higher number of BGCs and inset A-2 highlights that Alphaproteobacteria encode higher number of hserlactones in their genomes. (B) Principal coordinate plot based on BGC profile dissimilarity of genomes of plant isolates. Ellipses show the parametric smallest area around the mean that contains 80% of the probability mass of each genus. Statistical significance was calculated based on PERMANOVA. Taxonomic groups occupy distinct regions of the biosynthetic space as indicated by distinct clusters of members of each genus. Data have been shown for the top 7 abundant genera in plant isolates. (C) Forest plot displaying differential enrichment of BGC categories among the top 7 abundant genera in plant isolates. The $\log_{10}$ mean count per genome (dots) and 95% confidence interval (upper and lower bars) are shown only for differentially abundant (negative binomial; $P < 0.05$) BGC categories relative to Bacillus_A.

plant and soil environments. The *D*-test of Fritz and Purvis (21) showed that all BGC classes display phylogenetic signals among plant isolates (Fig. 3A-1). The *D*-test statistic, a measure of the strength of phylogenetic signals, ranged from 0.06 to 0.46 (average *D*-value for all BGCs = 0.22; $P_{random}$ <0.05), indicating that the strength of phylogenetic signals differed for the different BGC classes (Fig. 3A-1). We also calculated Pagel's λ, a measure of phylogenetic signal on continuous traits data, and found that λ values range from 0.80 to 0.99 (average λ value for all BGCs = 0.89; likelihood test $P < 0.00001$) (Table S19). These suggest that all BGCs display phylogenetic signal in plant isolates data set.

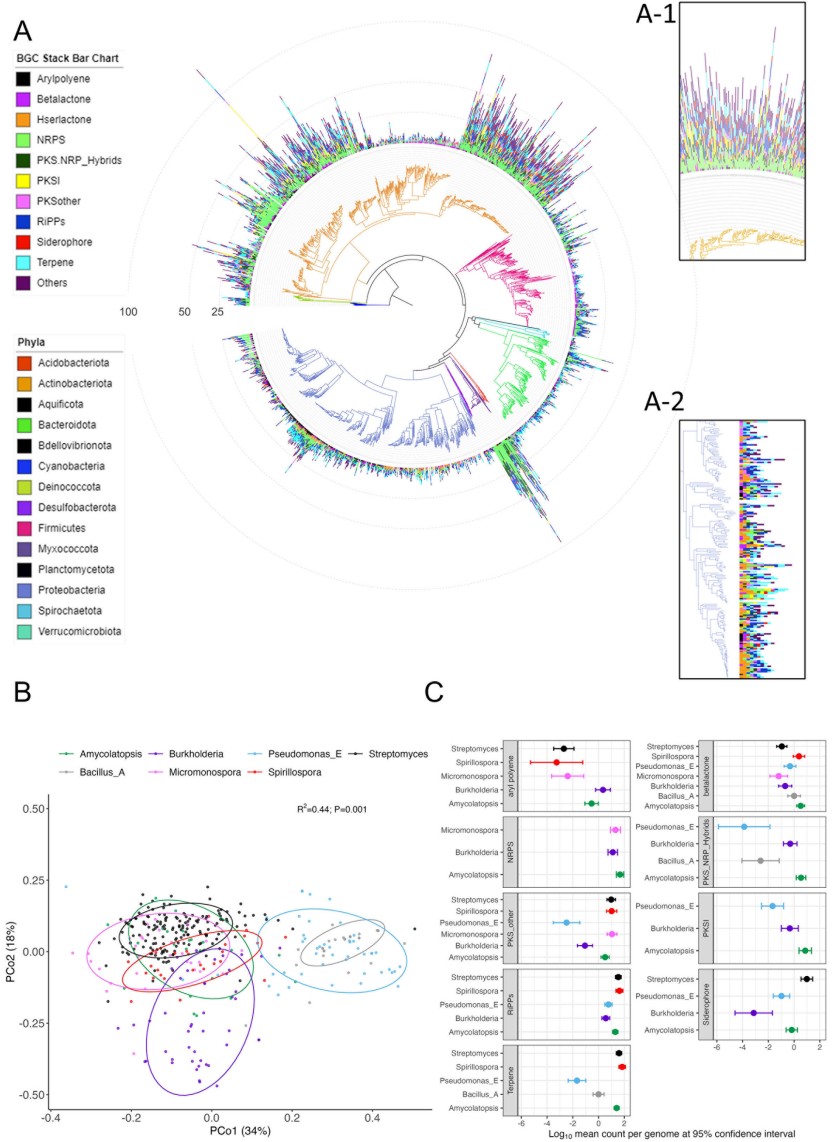

FIG 2 Phylogenetic distribution of BGCs in soil isolates. (A) Phylogenetic tree displaying the distribution of BGCs among members of soil isolates. Stacked bars in the outer rings indicate the count of different BGCs (shown in different colors) encoded by genomes of individual members. Taxonomic information (at the class level) is shown in colors of the phylogenetic tree branches. Inset A-1 highlights that Actinomycetes phylum possess overall higher number of BGCs and inset A-2 highlights that Alphaproteobacteria lacking characteristic presence of hserlactones compared to that of plant isolates. (B) Principal coordinate plot based on BGC profile dissimilarity of genomes of soil isolates. Ellipses show the parametric smallest area around the mean that contains 80% of the probability mass of each genus. Statistical significance was calculated based on PERMANOVA. Taxonomic groups occupy distinct regions of the biosynthetic space as indicated by distinct clusters of members of each genus. Data have been shown for the top 7 abundant genera in soil isolates. (C) Forest plot displaying differential enrichment of BGC categories among the top 7 abundant genera in soil isolates. The $\log_{10}$ mean count per genome (dots) and 95% confidence interval (upper and lower bars) are shown only for differentially abundant (negative binomial; $P < 0.05$) BGC categories relative to Amycolatopsis.

Furthermore, we applied the consenTRAIT approach (52) to estimate their mean genetic depth ($\tau_D$) of conservation. The consenTRAIT results demonstrated that BGCs are conserved in plant isolates, with a mean genetic depth ranging from 0.016 to 0.036 (average $\tau_D = 0.0229$) ($P < 0.05$) (Table 1). Taken together, the $D$-test and consenTRAIT

results established that BGCs are phylogenetically conserved among the phytobiomes. We applied all three approaches in plant MAGs and found that the BGCs are indeed phylogenetically conserved (average $D$-value for all BGCs = 0.77; $P_{random}$ <0.05), albeit the strength of the conservation was slightly weaker compared to plant isolates (Fig. 3A). The mean genetic depth ($\tau_D$) of the conservation of BGCs ranged from 0.038 to 0.069 (average $\tau_D$ = 0.0522) in the plant MAGs, which was similar to that in the plant isolates (Table 1).

## Terpene and aryl polyenes display stronger phylogenetic signals in phytobiomes

Noticeably, among the eleven BGC classes tested, the terpenes and aryl polyene classes had the strongest phylogenetic signals ($D$ = 0.06 and 0.11, respectively; $P_{random}$ <0.05) in the plant isolates. This pattern of terpene and aryl polyene classes with the strongest phylogenetic signal was also present in plant MAGs ($D$ = 0.66 and 0.55, respectively; $P_{random}$ <0.05) but not in the soil isolates and MAGs (Fig. 3A). Visualization of the distribution of strongly (terpene and aryl polyene BGC classes) and weakly conserved BGC classes (PKS-NRP hybrids and beta-lactone) in the phylogenetic tree of both plant isolates and plant MAGs further confirmed our claim (Fig. 3B and Fig. S2A for plant isolates and plant MAGs, respectively). These results support that terpene and aryl polyene BGCs are strongly phylogenetically conserved among phytobiomes. Between these two, terpene BGCs were found to be more predominant among phytobiomes (74.6% and 34.5% of the members in plant isolates and 29.3% and 24.9% of the members in plant MAGs possess terpene and aryl polyene biosynthetic capacity, respectively).

## Terpene-related sequences are phylogenetically clustered and mostly uncharacterized in phytobiomes

To improve our understanding of the phylogenetic conservation of terpenes, we further studied (i) the relationships among terpene-related sequences (GCFs) from different bacterial classes and (ii) the functional potential of these sequences. The sequence similarity network of terpene-related sequences in the plant isolates and plant MAGs revealed that only a few of terpene BGCs (12 out of 180 and 1 out of 44 in plant isolates and plant MAGs, respectively) had reference annotations (from the MIBiG database). The largest networks, especially those of Alphaproteobacteria, Bacilli, and Bacteroidia, did not harbor any of the previously well-characterized terpene BGCs (Fig. 4A). None of the clusters from Bacilli and Bacteroidia contain any previously characterized terpene BGCs, as further described below. Hence, members of these classes have the functional potential to synthesize novel terpenoids. Interestingly, in the well-studied Actinomycetia, previously characterized terpene-related sequences were indeed present in the largest clusters but not in smaller clusters (Fig. 4A). On the other hand, the terpene BGCs from plant MAGs were mostly singletons reflecting the possibility of diverse and novel molecules being encoded by these genomes (Fig. S2B).

This widespread occurrence of uncharacterized terpene clusters prompted us to investigate the domains of their core genes to understand their relationship with each other and their distribution in the different microbial clades. As expected, squalene/phytoene synthase (SQS-PSY), which is involved in the first step of tri- and tetraterpene biosynthesis, was the most widely distributed (~83% of members) (Fig. 4B). Interestingly, SQS-PSY is absent in specific clades of Bacilli (e.g., *B. altitudinis*, *B. safensis*, and *B. cereus*, among others). Bacilli, in general, did not possess carotenoid BGCs, while the majority possessed Squalene Hopene Cyclase (SHC) clusters. The well-characterized Actinomycetia (*Streptomyces* spp.) harbored all core terpene domains, indicating their potential to produce diverse types of tri- and tetraterpenes. Both sequence similarity network and domain analyses suggested that Alphaproteobacteria potentially can produce distinct tri- and/or tetra-terpenes via their unique SQS-PSY domains (Fig. 4A and B; Fig. S3). In summary, based on sequence similarity networks and the domain distribution of terpene

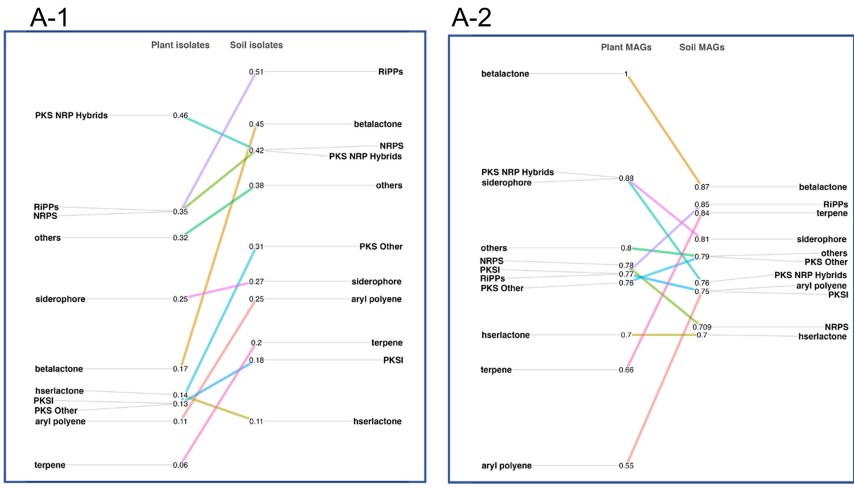

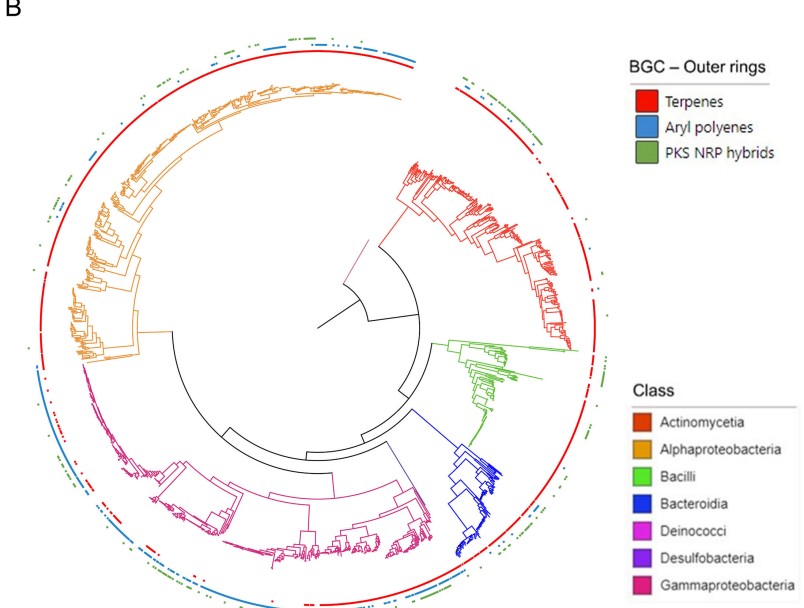

**FIG 3** Phylogenetic conservation of Biosynthetic gene clusters. (A) Slope chart displaying the strength of phylogenetic conservation in isolates (A-1) and MAGs (A-2) data sets. A lower $D$-value indicates stronger phylogenetic conservation. The D-statistics were statistically significant ($P_{random}$ <0.05 and $P_{Brownian}$ <0.05) for all BGC classes, except for beta-lactone in soil isolates and plant MAGs data sets. (B) Phylogenetic tree showing presence/absence pattern of terpenes, aryl polyenes, and PKS-NRP hybrids in plant isolates genomes. Stronger phylogenetic conservation of terpenes and aryl polyenes compared to PKS-NRP hybrids are shown by their three respective outer rings (red, blue, and green, respectively). The taxonomic classes of the plant isolates genomes are shown in different colors of the phylogenetic tree branches.

clusters, it is evident that phytobiomes are rich in potentially novel biosynthetic genes in previously understudied clades.

## DISCUSSION

We investigated the relationship between microbial phylogeny and the biosynthetic potential of specialized metabolites in a worldwide collection of isolates and MAGs from diverse plant hosts. This is the first attempt to investigate the biosynthetic potential of phytobiomes using such global-scale resources, expanding on previous reports based on

A

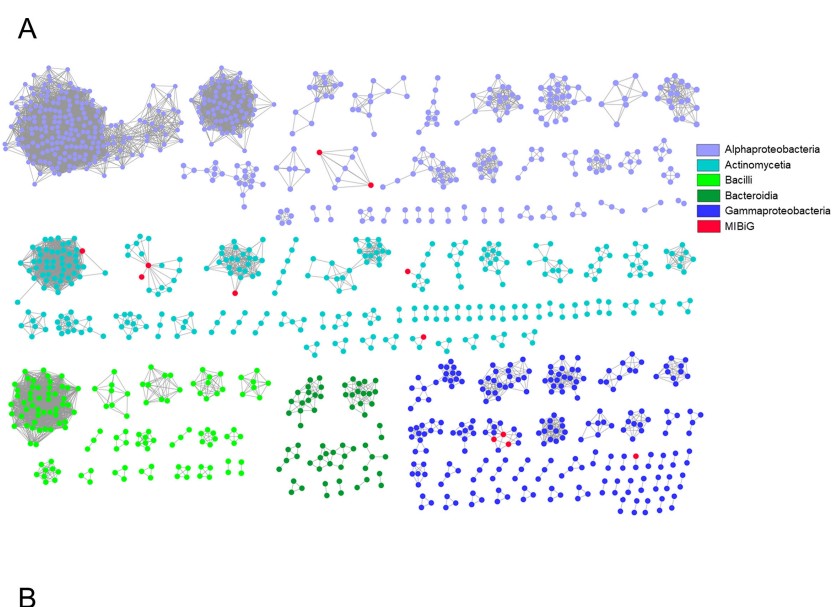

B

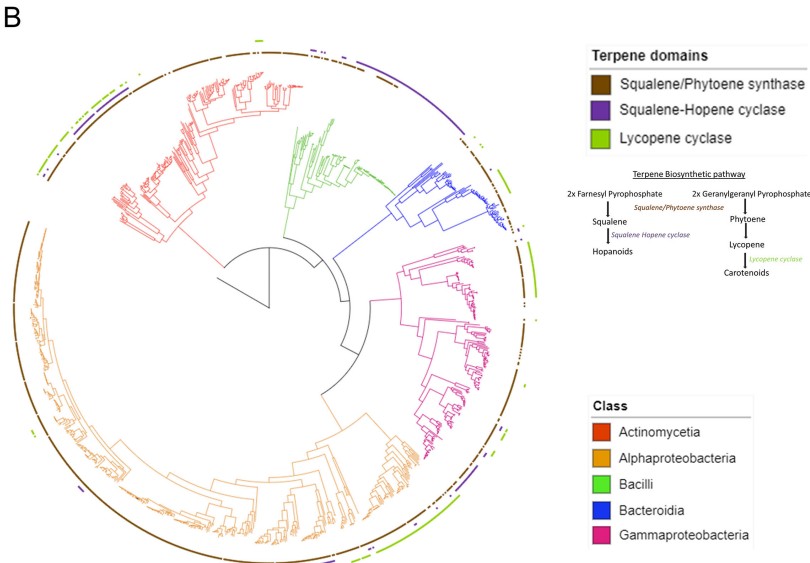

**FIG 4** Sequence similarity network and domain distribution of terpenes in plant isolates. (A) Sequence similarity network of terpenes GCFs found in the plant isolates genomes (singletons are not shown). Each node represents individual terpene BGC, colored according to taxonomy. Nodes with existing reference (MIBiG) are highlighted in red. (B) Phylogenetic tree showing presence/absence pattern of terpene biosynthetic domains in plant isolates genomes. Distribution patterns of Squalene/Phytoene synthase, Squalene-hopene cyclase, and Lycopene cyclase are shown in separate colors of the outer rings (brown, violet, and green, respectively). The taxonomic level (class) of plant isolates genomes is shown in different branch colors of the phylogenetic tree. The terpene biosynthetic pathway for hopanoid and carotenoid biosynthesis is drawn with major enzyme involved in it.

either MAGs or isolates from specific plant hosts or microbial taxa (32, 53–55). This resource would help in generating experimentally verifiable hypotheses on how specific microbial groups can contribute to chemical ecology in different niches and environments. The framework for generating such hypotheses is reported here, and this resource could be used to reliably reveal conserved patterns of BGCs in both isolates and MAGs based on the choice of retaining only high-quality and nonredundant genomes. However, owing to the inherent differences in genomic characteristics and in BGC contents between isolates and MAGs, we recommend separate use of the two data sets.

The overall approach used here could be extended to other ecologically relevant microbial traits, such as antibiotic resistance (ARDB) (56), biogeochemical cycling of nutrients (57), and those involving carbohydrate active enzymes (CAZymes) (58).

Plants and microbes have co-evolved for millennia, which has led to their traits being highly influenced by both plant–microbe and microbe–microbe interactions (59, 60). The differential phylogenetic signal strength of several BGC classes in the phytobiomes shown in this study is suggestive of a strong plant host selection pressure in shaping the evolution of these traits in phytobiomes. Since selection pressures act distinctly on different BGC classes (13, 61), the evolutionary processes leading to differences in their strength of conservation under host influence may be somewhat different and independent. The well-recognized role of specialized metabolites in microbe–microbe communications, signaling, and antagonism also suggests that microbe–microbe and plant–microbe interaction components together contribute to the complex eco-evolution of BGCs in phytobiomes (62, 63).

Considering BGCs as microbial traits, this approach presents an opportunity to measure and compare their complexity with that of other microbial traits. The genetic depth of conservation of BGCs ($\tau_D = 0.022$–$0.053$) is equivalent to that of other complex microbial traits, such as methanogenesis ($\tau_D = 0.042$) and sulfate reduction ($\tau_D = 0.039$) (64). Since trait complexity is correlated with the number of genes underlying a particular trait (65), the highly diverse components of metabolic machinery, such as enzymes, transporters, regulators, and accessory proteins, together contribute to establish BGCs as a complex microbial trait (66).

The phylogenetic signal strength of microbial traits provides clues to the mode of their inheritance (vertical/horizontal inheritance) among community members in an ecosystem. Terpene and aryl polyene BGC classes displayed stronger conservation than did other BGC classes in the phytobiomes, suggesting that vertical inheritance has a strong influence on their distribution. However, we did not specifically investigate this in this work but provide testable hypothesis for such patterns of inheritance among microbes. One explanation of their similar strength of conservation could be the result of their functionally convergent roles in antioxidative processes, which is imperative for bacterial colonization of plant roots (19, 67, 68). The strong conservation of terpenes in phytobiomes coincides with the phylogenetic conservation of terpenes in members of the plant kingdom (69). Furthermore, a recent report suggested that members of different bacterial clades respond differently toward terpenes secreted from plant roots (5). Taken together, these findings will provide important clues to the eco-evolutionary trajectories of terpenes in this "holobionts" entity (70).

Phylogenomic approaches can facilitate discoveries of molecules with potentially novel biochemical properties through rationally guided hypothesis generation (13). Here, we pinpoint such microbial clades to search for potentially novel terpene classes. First, Bacteroidia and Bacilli, the major abundant taxa in phytobiomes, harbor uncharacterized terpenes, directing future studies to focus on these clades to search for functionally novel terpenes. Second, specific members of Alphaproteobacteria form a distinct lineage of SQS/PSY domains, indicating that these domain-containing enzymes could possess novel biochemical properties. We also noted an instance of clade-specific biochemical adaptations that could shed light on the evolutionary mechanisms of terpenes in phytobiomes. For instance, the absence of SQS/PSY but the presence of SHC in specific clades of Bacilli could suggest that throughout the course of coevolution with plants, they either have evolved to take up squalene from other phytobiome members or have lost the capacity to produce hopanoids.

In conclusion, this study presents a novel approach through which chemical and microbial ecology is combined via a phylogenomics framework. One potential application of this approach, though investigated little here, is in understanding the coevolution of hosts and microbiomes through specialized metabolites. This can be further investigated in subsequent studies. The second application, as demonstrated here with terpenes, is in rationally guiding the discovery of novel metabolites classes that can

potentially be produced by distinct clades of host-associated microbiomes. This could be particularly helpful in determining the roles of widespread cryptic metabolic pathways for natural product discovery.

## ACKNOWLEDGMENTS

The project is supported by the National Research Foundation, Prime Minister's Office, Singapore, with Grant ID NRF-CRP16-2015-04, titled as "Novel integrated agrotechnologies, plant nutrients and microbials for improved production of green leafy vegetables in Singapore." A.M. is a recipient of SCELSE-NUS Graduate fellowship. The authors thank the authors whose data contributed to this analysis. The authors also acknowledge the use of ChatGPT for grammatical corrections in this manuscript.

A.M. and H.T. contributed equally to the overall manuscript. The author orders were determined alphabetically. A.M. contributed to conceiving, phylogenomics framework development, statistical analyses, manuscript drafting, and review. H.T. contributed to conceiving, network analyses, biosynthetic gene clusters analyses, manuscript drafting, and review. A.B. provided inputs for statistical methods. S.P. contributed to developing framework and reviewing the manuscript. S.S. provided inputs for conceptual development, manuscript drafting, and review. All authors read and approved the final manuscript.

The authors declare no competing interests.

## AUTHOR AFFILIATIONS

[1]Department of Biological Sciences, National University of Singapore, Singapore, Singapore

[2]Singapore Centre for Environmental Life Sciences Engineering, National University of Singapore, Singapore, Singapore

[3]NUS Environmental Research Institute, National University of Singapore, Singapore, Singapore

## AUTHOR ORCIDs

Arijit Mukherjee  http://orcid.org/0000-0003-3329-3535

## FUNDING

| Funder | Grant(s) | Author(s) |
| --- | --- | --- |
| National Research Foundation Singapore (NRF) | NRF-CRP16-2015-04 | Sanjay Swarup |

## AUTHOR CONTRIBUTIONS

Arijit Mukherjee, Conceptualization, Data curation, Formal analysis, Methodology, Visualization, Writing – original draft, Writing – review and editing | Hitesh Tikariha, Conceptualization, Data curation, Formal analysis, Methodology, Software, Visualization, Writing – original draft, Writing – review and editing | Aditya Bandla, Methodology | Shruti Pavagadhi, Project administration, Writing – review and editing | Sanjay Swarup, Conceptualization, Funding acquisition, Supervision, Writing – original draft, Writing – review and editing

## DATA AVAILABILITY STATEMENT

Data used in the study were collected from public repositories and can be retrieved by using the information provided in the Materials and Methods. All codes to reproduce the analysis and generate figures of this manuscript have been deposited in GitHub. Software versions and nondefault parameters used have been appropriately specified where required.

## ADDITIONAL FILES

The following material is available online.

### Supplemental Material

**Supplemental figures (mSystems00387-23-S0001.pdf).** Figures S1 to S3.
**Supplemental tables (mSystems00387-23-S0002.xlsx).** Tables S1 to S19.

### Open Peer Review

**PEER REVIEW HISTORY (review-history.pdf).** An accounting of the reviewer comments and feedback.

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
