## [Reviewer comments · mSystems]

Global analyses of biosynthetic gene clusters in phytobiomes reveal strong phylogenetic conservation of terpenes and aryl polyenes

Arijit Mukherjee, Hitesh Tikariha, Aditya Bandla, Shruti Pavagadhi, and Sanjay Swarup

Corresponding Author(s): Sanjay Swarup, National University of Singapore

Review Timeline:

Submission Date:	April 21, 2023
Editorial Decision:	May 15, 2023
Revision Received:	May 18, 2023
Accepted:	May 30, 2023

Editor: Aurélie Deveau

Reviewer(s): The reviewers have opted to remain anonymous.

Transaction Report:

DOI: <https://doi.org/10.1128/msystems.00387-23>

May 15, 2023

Dr. Sanjay Swarup
National University of Singapore
Singapore
Singapore

Re: mSystems00387-23 (Global analyses of biosynthetic gene clusters in phytobiomes reveal strong phylogenetic conservation of terpenes and aryl polyenes)

Dear Dr. Sanjay Swarup:

Thank you for submitting your manuscript to mSystems. We have completed our review and I am pleased to inform you that, in principle, we expect to accept it for publication in mSystems. However, acceptance will not be final until you have adequately addressed the reviewer comments.

Both reviewers appreciated the efforts made to answer their comments and found that the results greatly improve the manuscript. However, one reviewer suggested some minor improvements to enhance the readability of the manuscript. Please make these few corrections.

Preparing Revision Guidelines

Please return the manuscript within 60 days; if you cannot complete the modification within this time period, please contact me. If you do not wish to modify the manuscript and prefer to submit it to another journal, please notify me of your decision immediately so that the manuscript may be formally withdrawn from consideration by mSystems.

Sincerely,

Aur lie Deveau

Editor, mSystems

Journals Department
Reviewer comments:

Reviewer #1 (Comments for the Author):

The paper by Mukherejee et al. describes a genomic analysis of bacteria from phytobiomes and the link between BGCs and their conservation in these systems with an emphasis on the presence of terpenes and aryl polyenes. The article was well written and has even improved after revision.

Reviewer #2 (Comments for the Author):

I acknowledge the effort put in by the authors to address the initial review comments, and their new analyses have considerably reinforced their initial conclusion. The manuscript could still benefit from some minor refinements. I provide some examples below

Minor comments:

P6 line 164: For "JGI" spell out what this abbreviation and give website or citation to specific database that was used for analysis.

P8 line 250: "To obtain more" should be "To obtain a more..."

P9 line 254: "to reconstruct best..." should read "to reconstruct the best..."

P9 line 274: Does "bray" refer to Bray-Curtis dissimilarity? please define and add a citation.

P11 line 343: It's not clear to the reviewer what "processed" specifically refers to? Are you just saying that taxonomy was assigned to each genome? If so how?

P12 line 358: Again, not clear what "80% of the alignment has gaps" specifically is referring to? Please clarify. Same line is "BMGE" abbreviation defined somewhere?

P12 line 377: Somewhere in the document wording needs to be made obvious---by "plant isolates" you mean "bacterial isolates" from plants or soil.

P14 line 419: "Streptomces" should be "Streptomyces" and "sp." should not be italicized.

P14 line 425: Should Mesorhizobium, Bradyrhizobium, Rhizobium be italicized?

P18 line 539: Please define what is meant by GCFs?

I acknowledge the effort put in by the authors to address the initial review comments, and their new analyses have considerably reinforced their initial conclusion. The manuscript could still benefit from some minor refinements. I provide some examples below.

Minor comments:

P6 line 164: For “JGI” spell out what this abbreviation and give website or citation to specific database that was used for analysis.

P8 line 250: “To obtain more” should be “To obtain a more...”

P9 line 254: “to reconstruct best...” should read “to reconstruct the best...”

P9 line 274: Does “bray” refer to Bray-Curtis dissimilarity? please define and add a citation.

P11 line 343: It’s not clear to the reviewer what “processed” specifically refers to? Are you just saying that taxonomy was assigned to each genome? If so how?

P12 line 358: Again, not clear what “80% of the alignment has gaps” specifically is referring to? Please clarify. Same line is “BMGE” abbreviation defined somewhere?

P12 line 377: Somewhere in the document wording needs to be made obvious---by "plant isolates" you mean "bacterial isolates" from plants or soil.

P14 line 419: “*Streptomces*” should be “*Streptomyces*” and “sp.” should not be italicized.

P14 line 425: Should *Mesorhizobium*, *Bradyrhizobium*, *Rhizobium* be italicized?

P18 line 539: Please define what is meant by GCFs?

RESPONSE TO REVIEWERS (mSystems00387-23R1)

Global analyses of biosynthetic gene clusters in phytobiomes reveal strong phylogenetic conservation of terpenes and aryl polyenes

Authors:

Arijit Mukherjee, Hitesh Tikariha, Aditya Bandla, Shruti Pavagadhi, and Sanjay Swarup

The responses to the reviewer's comments are highlighted in blue and are provided on a point-by-point basis.

Reviewer #1 (Comments for the Author):

The paper by Mukherejee et al. describes a genomic analysis of bacteria from phytobiomes and the link between BGCs and their conservation in these systems with an emphasis on the presence of terpenes and aryl polyenes. The article was well written and has even improved after revision.

Response:

We thank the reviewer for constructive suggestions for improving the manuscript.

Reviewer #2 (Comments for the Author):

I acknowledge the effort put in by the authors to address the initial review comments, and their new analyses have considerably reinforced their initial conclusion. The manuscript could still benefit from some minor refinements.

Minor comments:

Response:

Point by point responses for each of the comments are provided below-

P6 line 164: For "JGI" spell out what this abbreviation and give website or citation to specific database that was used for analysis.

We have now included the abbreviation and website for JGI in this revised manuscript. See line 164 in the revised manuscript.

P8 line 250: "To obtain more" should be "To obtain a more..."

We have now incorporated this change in the revised manuscript. Please see line 247 in this revised manuscript.

P9 line254: "to reconstruct best..." should read "to reconstruct the best..."

We have now incorporated this change in the revised manuscript. Please see line 251 in this revised manuscript.

P9 line 274: Does "bray" refer to Bray-Curtis dissimilarity? please define and add a citation.

Yes, 'bray' refers to the Bray-Curtis dissimilarity. We have now added a citation for this dissimilarity measure. See line 266 in the revised manuscript.

P11 line 343: It's not clear to the reviewer what "processed" specifically refers to? Are you just saying that taxonomy was assigned to each genome? If so how?

Indeed, the previous version implied that we have processed the network sequences to assign taxonomy. However, we only overlaid the taxonomic information of the genomes for the sequence networks. Therefore, we have now rephrased the sentence as -

"The network file of terpenes containing GCF information of both plant isolates and plant MAGs obtained from BiG-SCAPE were overlaid with taxonomic information (at the class level) of the genomes."

See line 336-340 in the revised manuscript.

P12 line 358: Again, not clear what "80% of the alignment has gaps" specifically is referring to? Please clarify. Same line is "BMGE" abbreviation defined somewhere?

While aligning amino acid sequences, we generally encounter some gaps since the sequences are not identical. Since in this case there are large variations among the sequences, we observed an overall 80% gap among the aligned sequences.

We have now added the abbreviation of BMGE (Block Mapping and Gathering with Entropy). See line 250 in the revised manuscript.

P12 line377: Somewhere in the document wording needs to be made obvious---by "plant isolates" you mean "bacterial isolates" from plants or soil.

We have now made it obvious that by 'plant isolates' we mean bacterial isolates from plants. Same is indicated for soil isolates, plant MAGs, and soil MAGs. See line 130-135 in the revised manuscript-

"First, we collated phytobiome datasets from cultured bacterial isolates (referred as 'plant isolates') and metagenome-assembled genomes (referred as 'plant MAGs') and included soil-associated cultured bacterial isolates (referred as 'soil isolates') and MAGs (referred as 'soil MAGs') as reference."

P14 line 419: "Streptomces" should be "Streptomyces" and "sp." should not be italicized.

We have incorporated the change as suggested. See line 415 in the revised manuscript.

P14 line 425: Should Mesorhizobium, Bradyrhizobium, Rhizobium be italicized?

We have incorporated the change as suggested. See line 420-421 in the revised manuscript.

P18 line 539: Please define what is meant by GCFs?

We have previously mentioned the meaning of GCF i.e. Gene Cluster Family in this manuscript. See line 228 in this revised submission.

May 30, 2023

Dr. Sanjay Swarup
National University of Singapore
Singapore
Singapore

Re: mSystems00387-23R1 (Global analyses of biosynthetic gene clusters in phytobiomes reveal strong phylogenetic conservation of terpenes and aryl polyenes)

Dear Dr. Sanjay Swarup:

Your manuscript has been accepted, and I am forwarding it to the ASM Journals Department for publication. For your reference, ASM Journals' address is given below. Before it can be scheduled for publication, your manuscript will be checked by the mSystems production staff to make sure that all elements meet the technical requirements for publication. They will contact you if anything needs to be revised before copyediting and production can begin. Otherwise, you will be notified when your proofs are ready to be viewed.

If you would like to submit a potential Featured Image, please email a file and a short legend to msystems@asmusa.org. Please note that we can only consider images that (i) the authors created or own and (ii) have not been previously published. By submitting, you agree that the image can be used under the same terms as the published article. File requirements: square dimensions (4" x 4"), 300 dpi resolution, RGB colorspace, TIF file format.

We recognize that the video files can become quite large, and so to avoid quality loss ASM suggests sending the video file via <https://www.wetransfer.com/>. When you have a final version of the video and the still ready to share, please send it to mSystems staff at msystems@asmusa.org.

Sincerely,

Aurélie Deveau
Editor, mSystems

Journals Department
E-mail: mSystems@asmusa.org